# Defogging Learning Based on an Improved DeepLabV3+ Model for Accurate Foggy Forest Fire Segmentation

Tao Liu [1], Wenjing Chen [1], Xufeng Lin [1], Yunjie Mu [1], Jiating Huang [1], Demin Gao [1] and Jiang Xu [2,*]

1    College of Information Science and Technology, Nanjing Forestry University, Nanjing 210037, China
2    School of Computer Science and Engineering, Changshu Institute of Technology, Changshu 215500, China
*    Correspondence: xjcs@cslg.edu.cn; Tel.: +86-155-0152-1663

**Abstract:** In recent years, the utilization of deep learning for forest fire detection has yielded favorable outcomes. Nevertheless, the accurate segmentation of forest fires in foggy surroundings with limited visibility remains a formidable obstacle. To overcome this challenge, a collaborative defogging learning framework, known as Defog DeepLabV3+, predicated on an enhanced DeepLabV3+ model is presented. Improved learning and precise flame segmentation are accomplished by merging the defogging features produced by the defogging branch in the input image. Furthermore, dual fusion attention residual feature attention (DARA) is proposed to enhance the extraction of flame-related features. The FFLAD dataset was developed given the scarcity of specifically tailored datasets for flame recognition in foggy environments. The experimental findings attest to the efficacy of our model, with a Mean Precision Accuracy (mPA) of 94.26%, a mean recall (mRecall) of 94.04%, and a mean intersection over union (mIoU) of 89.51%. These results demonstrate improvements of 2.99%, 3.89%, and 5.22% respectively. The findings reveal that the suggested model exhibits exceptional accuracy in foggy conditions, surpassing other existing models across all evaluation metrics.

**Keywords:** forest fire segmentation; deep learning; defogging





## 1. Introduction

Forests are one of the most vital ecosystems on Earth, providing essential ecosystem services such as oxygen and ecological balance [1], as well as serving as habitat and food sources for animals and plants. In addition, forests are important resources for society due to mitigating climate change and reducing carbon dioxide emissions [2]. However, forest fires represent a significant risk to these ecosystems, causing extensive damage and environmental losses globally. With over 35,000 forest fires occurring annually, millions of hectares of forest are consumed, leading to significant economic and environmental impacts [3–5]. Therefore, effective monitoring and evaluation of forest fires are crucial for us to minimize the loss of natural resources.

Traditional methods for detecting forest fires, such as manual monitoring, thermal imaging, and radio detection, often have low identification rates and require significant time and effort [6]. While deep neural networks have made substantial advances in forest fire detection [7], detecting forest fires in hazy images remains a challenging task. Foggy conditions can cause images to become blurry, making it more difficult to distinguish the forms, ranges, and spectra of fires [8,9]. More research is needed to create new algorithms and strategies to increase the accuracy and usefulness of forest fire recognition in foggy forests.

Defogging image processing has seen significant advancements in recent years, driven in part by the use of the physical scattering model to formulate the fogging process [10,11]. The model describes the interaction of light with atmospheric particles to simulate the scattering and absorption of light in foggy conditions.

Two types of single-picture defogging techniques have emerged in the last decade. The first category is based on foggy priors, such as dark channels [12], color attenuation [13],

and haze lines [14]. However, these methods frequently rely on complicated statistical hypotheses and physical models that may not precisely assess physical characteristics and apply to various situation [15]. The second group includes techniques based on deep learning, including AOD-Net [16], which recovers images by reformulating the physical scattering mode, end-to-end networks designed by Cai et al. [17] and Chen et al. [18], which do not rely on foggy priors, and the DCPDN model suggested by Zhang et al. [19]. Dong et al. [20] also incorporated enhancement strategies into their network to gradually recover clean images. These methods have demonstrated remarkable performance in defogging.

More and more deep learning techniques are being used to solve the issue of forest fires in hazy photos. For example, Yang et al. [21] designed a smoke detection model that leverages dark channel-assisted hybrid attention, while Merve Balki Tas et al. [22] created a smoke detection idea specifically for foggy wildfires. Huang et al. [23] also designed a GXTD detection model with a defogging function. These approaches, however, may have drawbacks such as indistinct edges, poor adaptation to fluctuating fog concentrations, and a high false alarm rate. On the other hand, there have been effective solutions developed for defogging segmentation of other objects. For instance, IRDCLNet provides a fog segmentation solution for ships that uses an interference reduction module and a dynamic contour learning module [24]. Another illustration is the work of Zhu et al., which enhances the defogging technique by using a dual attention mechanism and an SOS acceleration module [25]. He et al. [26] also propose using residuals and attention mechanisms to address the problem of different fog concentrations. These methods have shown promising results in their respective domains. It is also worth mentioning Chen et al.'s semi-supervised joint defogging learning framework, which effectively identifies vehicles in foggy weather by fusing feature layers [27]. This approach has demonstrated superior performance in accurately recognizing vehicles in foggy conditions.

In our study, we present an improved joint defogging semantic segmentation model named Defog DeepLabV3+. It is built upon DeepLabV3+ [28] and is designed to enhance the stability and accuracy of the segmentation of forest fires. A branch for dehazing and a branch for segmenting forest fires make up the suggested model. The dehazing branch uses pyramid feature enhancement to further hone the dehazed image and a two-stage feature extraction approach to improve the quality of the input image. On the other hand, the forest fire segmentation branch utilizes clean image dehazing features acquired from the dehazing branch. The segmentation branch integrates Atrous Spatial Pyramid Pooling (ASPP) [29] to gather contextual data at multiple scales, and an attention mechanism to increase the features' ability to discriminate, thus achieving better forest fire segmentation results. By sharing fused features, the model enhances the defogging and segmentation modules and aims to enhance the overall effectiveness of detecting forest fires in cloudy photos.

The following are the primary contributions of this paper:

- We design a smart attention mechanism module in our model, consisting of two independent branches that use different attention strategies. The module also uses residual structures to enhance the model's feature representation capabilities.
- Our model proposes a new training framework that unifies defogging and segmentation networks. The joint defogging learning framework preserves defogging features for forest fire segmentation, enabling the model to cope with poor visibility.
- Because no dataset exists specifically for foggy weather forest fire segmentation, we restructured existing benchmarks and created the Foggy FLAME Dataset (FFLAD). This dataset is intended to aid research in this area by serving as a baseline for evaluating the efficacy of defogging and segmentation models in forest fire detection under low visibility situations.
- Our experimental findings show that our Defog DeepLabV3+ model outperforms several existing CNN-based algorithms in the segmentation of foggy forest fire instances.

This work is organized as follows. Section 2 describes the experimental material and methods for detecting fogged forest fires, including the joint training framework of the

defogging network and the segmentation network. The experimental data and analyses are presented in Section 3. The debate is presented in Section 4. Section 5 concludes with some last thoughts.

## 2. Materials and Procedures

This paragraph describes the FFLAD dataset and production procedure. We also designed a joint training model for the defogging and segmentation networks called Defog DeepLabV3+. The first challenge addressed in this model is deep feature defogging. The second task is forest fire semantic segmentation, with an emphasis on improving forest fire segmentation accuracy.

### 2.1. Dataset

#### 2.1.1. Data Source

Because of the scarcity of fogged forest fire data and the scarcity of mature fogged forest fire datasets, we created a comprehensive dataset called FFLAD by selecting suitable forest fire detection datasets that were applicable for fogging. Our data sources include the FLAME dataset [30], the Bowfire dataset [31], and the ERA dataset [32]. The FLAME dataset consists of fire aerial images of combusting waste in pine forests in Arizona. The Bowfire dataset contains numerous negative samples that can be easily mistaken for flames, including forest fire images in various scenarios. We extracted forest fire scenes from multiple unmanned aerial vehicle (UAV) videos in the ERA dataset to construct our dataset. These high-quality datasets have been annotated by researchers and serve as a suitable foundation for our Fog Flame Dataset (FFLAD). Figure 1 showcases some representative images.

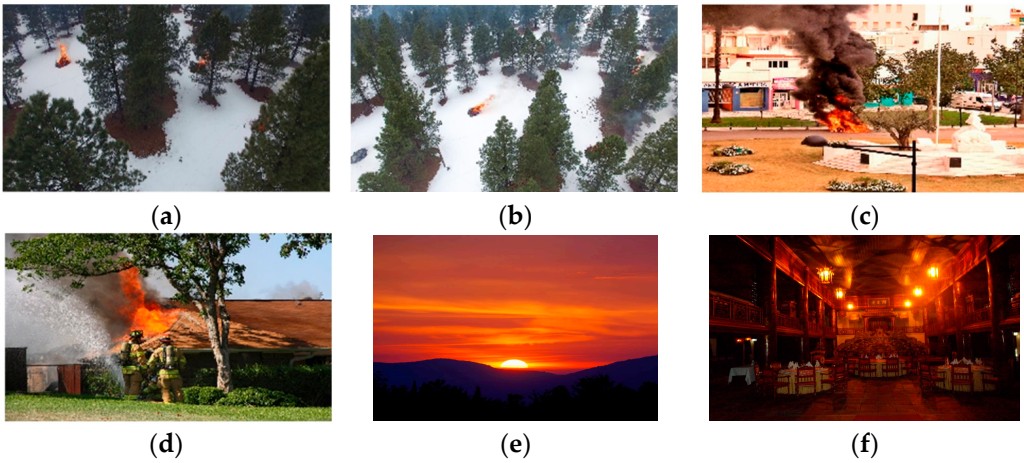

**Figure 1.** Images of possible forest fire. (**a**,**b**) Small target fire in the snow; (**c**) smoky larger fire; (**d**) dense area fire; (**e**,**f**) other objects similar to fire.

In addition, we obtained forest fire photos from Internet search engines as an additional test set to assess the generalization performance of our model. The selected fire images had diverse scenes and contained many interfering objects.

The dataset consisted of foggy and clear images in RGB format with a bit depth of 24. Each image had a resolution of $3840 \times 2160$ or $1024 \times 680$ pixels and a horizontal and vertical resolution of 96 dpi. The input image size for the model was set to $512 \times 512$ pixels. The label images were grayscale images with a bit depth of 8, where the label for the fire had a pixel value of 1. The size of the label images was consistent with that of the input images. The dataset had a size of 16.54 GB.

#### 2.1.2. Data Fogging and Enhancement Processing

For our experiments, we utilized the FLAME dataset and a subset of the Bowfire dataset as the original dataset and preprocessed them as needed. Specifically, we created a depth map for each clear map using the Monodepth2 self-supervised depth estimation

model [33]. In the case of a homogeneous atmosphere, the single-point transmittance (*t*) and the scene depth information (*d*) are related by the scattering factor in the atmosphere (*β*), according to the following equation [12,34]:

$$t(x) = e^{-\beta d(x)} \tag{1}$$

Equation (1) shows that the amount of radiation in the scene decreases exponentially as distance increases, where t is less than or equal to 1. To model the effect of atmospheric scattering, we used the equation described by reference [10,11]:

$$I(x) = J(x)t(x) + A(1 - t(x)) \tag{2}$$

$$J(x) = \frac{I(x) - A}{t(x)} + A \tag{3}$$

where *I(x)* represents the observed blurred image, *J(x)* is the scene brightness to be recovered, *t(x)* is the global atmospheric light and transmission maps, and A represents the atmospheric light.

$$I(x) = J(x)e^{-\beta d(x)} + A(1 - e^{-\beta d(x)}) \tag{4}$$

We generated the fog map using Equation (4), which is derived from Equations (1) and (2). In Equation (4), the constant A represents atmospheric light intensity and may vary depending on fog concentration and weather conditions. Typically, *A* falls within the range of 10 to 100 [11]. The fog maps are saved in ".jpg" format with the same dimensions as the original image. Ultimately, the FFLAD dataset comprises original images, two types of depth maps, fog maps, and label maps (Figure 2).

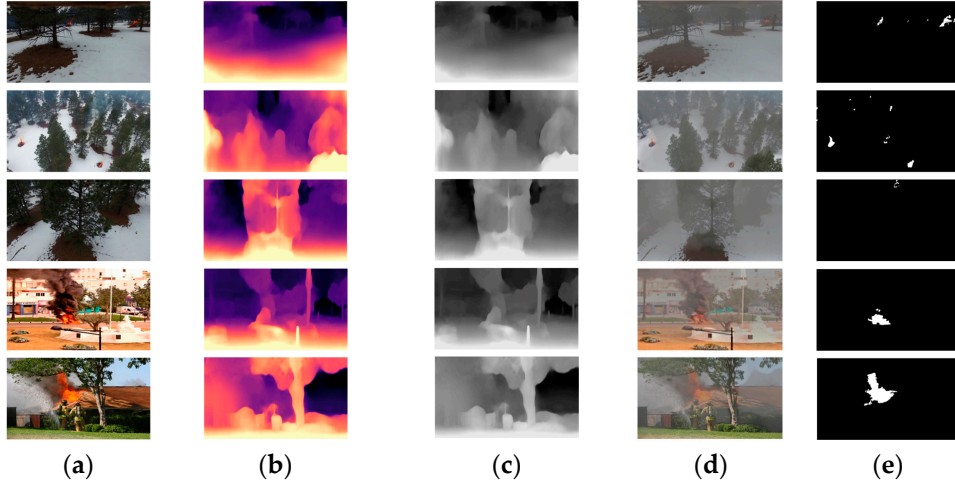

(a)　　　　　(b)　　　　　(c)　　　　　(d)　　　　　(e)

**Figure 2.** Fire samples of FFLAD: (**a**) clear images of wildfires; (**b**,**c**) depth map with three-channel and single-channel; (**d**) foggy images of wildfires; and (**e**) ground truth.

There were 5001 total photos in our dataset, which were split into three groups: a training set, a validation set, and a test set. In deep learning segmentation research on forest fires, the 80/10/10 data split is a commonly used practice, and we also adopted this approach. In addition, we conducted comparative experiments using data splits of 90/5/5, 70/15/15, and 70/20/10. The results showed that compared to the 80/10/10 data split, the other data splits led to overfitting, where the model performed well on the training set but poorly on the test set. The model performed well on both the training and validation sets, and its performance on the test set was similarly dependable, thanks to the very balanced 80/10/10 data split. These sets account for 80%, 10%, and 10% of the dataset, respectively (Table 1).

**Table 1.** The detailed information of the dataset FFLAD.

| Dataset | Number of Images | Proportion |
|---|---|---|
| Training set | 3564 | 80% |
| Validation | 445 | 10% |
| Testing set | 445 | 10% |
| Additional test set | 547 | - |
| Total | 5001 | 100% |

In addition to DeepLabV3+'s standard data enhancement techniques of random scaling, random cropping, and random color perturbation, our model employs additional data enhancement techniques such as random Gaussian blur, random rotation, and random color gamut translation. These techniques significantly enhance the model's performance and resilience, with only marginal overhead increases [35].

2.1.3. Model Training

The hyperparameter choices for applying DeepLabV3+ for image segmentation that impacts the model are the backbone network selection, batch size, image size, initial learning rate, decoder, number of iterations, and optimizer strategy. Based on DeepLabV3+ experiments, our hyperparameter selection strategy in the experiment is shown in Table 2:

**Table 2.** Training parameters of our model.

| Training Parameters | Details |
|---|---|
| Batch size | 16 |
| Initial learning rate | 0.01 |
| Decoder | ASPP |
| Number of iterations | 100 |
| Optimizer strategy | SGD |
| Image size | $512 \times 512$ |

*2.2. Defog DeepLabV3+ Architecture*

2.2.1. DeepLabV3+

The DeepLabV3+ network architecture is designed for semantic segmentation tasks. It has an encoder–decoder structure that leverages DeepLabV3 as the encoder to obtain detailed contextual information. As an improvement in the DeepLab series, DeepLabV3 inherits the overall sequential architecture of BackBone and DeepLab Head [29,36,37]. The Backbone is used to extract deep image features, and ResNet is one of the commonly used choices due to its strong feature extraction capability brought by the residual structure. The DeepLab Head is used to map the features extracted by the Backbone to pixel-level segmentation results, including the Atrous Spatial Pyramid Pooling (ASPP) module and decoder module. The ASPP module is a core component of DeepLabv3, which uses atrous convolutions with variable sampling rates to handle multi-scale features to get various scales' information. In addition, the ASPP module also includes a global pooling layer for handling global information of the entire image, thereby better capturing background information. The decoder module restores the feature map to the original image resolution through upsampling and deconvolution operations. However, ASPP itself introduces noise, and the large convolution kernels of atrous convolution cause the weight values to disperse, making the model sensitive to noise.

In DeepLabV3+, the decoder module is responsible for recovering object boundary information and extracting features from the encoder using an extended convolution of arbitrary resolution. Additionally, the encoder's backbone network employs a deeply separable convolution strategy [38] and an improved initial architecture [39] to reduce computational costs while maintaining accuracy.

The DeepLabV3+ network integrates the ASPP module in addition to the encoder–decoder structure to collect multi-scale contextual data. The ASPP module extracts features at various

scales while reducing the number of parameters by using atrous convolutions with various dilation rates [29]. This makes it possible for the network to efficiently gather both local and global contextual data, which are crucial for tasks requiring semantic segmentation.

Figure 3 describes the network architecture of DeepLabV3+. By combining the encoder–decoder structure, deeply separable convolutions, an improved initial architecture, and the ASPP module, DeepLabV3+ has displayed cutting-edge performance in semantic segmentation tasks on several benchmark datasets. This makes the architecture a powerful tool for various applications that require accurate semantic segmentation.

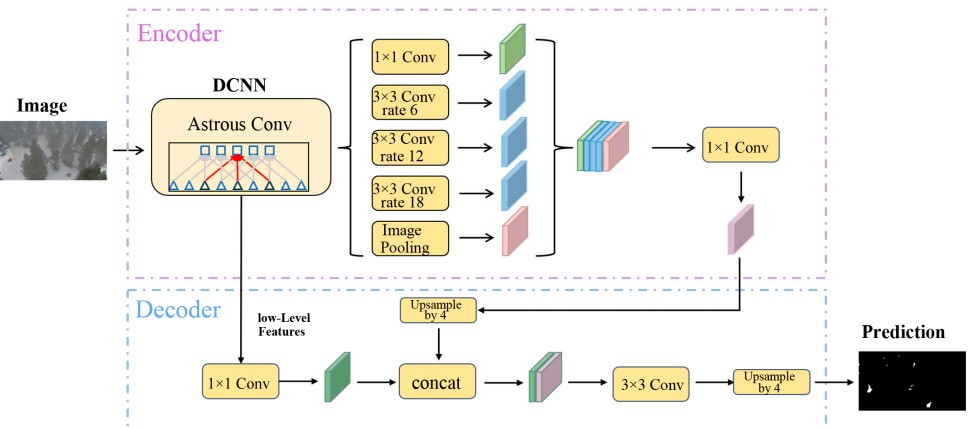

**Figure 3.** Model structure of DeepLabV3+.

For image segmentation studies of forest fires, effective advanced models have been proven to include UNet, DeepLabV3+, FCN, PSPNet, and the recently proposed MaskSU R-CNN [40]. However, UNet typically loses a lot of spatial information during downsampling, resulting in a small receptive field. Additionally, UNet only considers local context and has difficulty capturing global information, making it prone to overfitting. FCN's redundant fully convolutional structure has high memory consumption and results in blurred object boundaries in the segmentation output, without considering context [41]. While PSPNet to some extent overcomes the insufficiency of context information with its pyramid pooling structure, the shallow feature fusion is insufficient, resulting in poor segmentation details [42]. MaskSU R-CNN well inherits the high accuracy of MS R-CNN, but the model design is more complex and the scale is larger, with poor real-time performance [40]. In comparison, the DeepLabV3+ model can effectively enlarge the receptive field and collect the global information through atrous convolutions and the ASPP module, thereby performing excellently in segmentation details. DeepLabV3+ also controls the resolution of the segmentation output by changing the parameters of atrous convolution, thereby improving the segmentation speed while ensuring accuracy. Additionally, the DeepLabV3+ encoder–decoder structure achieves cross-layer feature passing, which is conducive to the joint design and use of the defogging framework.

Related studies tested the first 4 models' performance on the FLAME dataset [43], with UNet and DeepLabV3+ performing similarly with an accuracy of around 91%, while FCN and PSPNet achieved relatively lower accuracy at 85% and around 91% respectively. This once again validates the rationality of our model selection. Specifically, DeepLabV3+ achieved the highest mean IoU on the FLAME dataset, outperforming UNet, FCN, and PSPNet. The better performance can be attributed to DeepLabV3+'s ability to capture multi-scale contextual information through the ASPP module and effective decoder design. Compared to UNet's simple encoder–decoder architecture, DeepLabV3+ exploits richer semantic information from inputs through atrous separable convolutions and global context modeling, enabling more precise segmentation of flames and flame-like objects in forest fire images.

### 2.2.2. Defog DeepLabV3+

As described in Figure 4, our architecture comprises two main branches: the forest fire segmentation branch and the defogging branch. The collective feature sharing module (CFSM), which is shared by both branches during training, makes sure that the fog-free features produced by this module may be applied to either branch of our joint defogging learning architecture. In the fog removal branch, a two-stage recover module further processes the obtained depth information into dehazing feature and the pyramid enhancement block generates the final fog-free results. In the forest fire division branch, the deep feature module is composed of the designed fusion attention residual module and ASPP. ASPP is the key module designed in DeepLabV3+. During inference, only the CFSM and segmentation modules need to be executed. This architecture significantly improves the performance of forest fire segmentation in foggy weather without adding a computational burden during the inference phase.

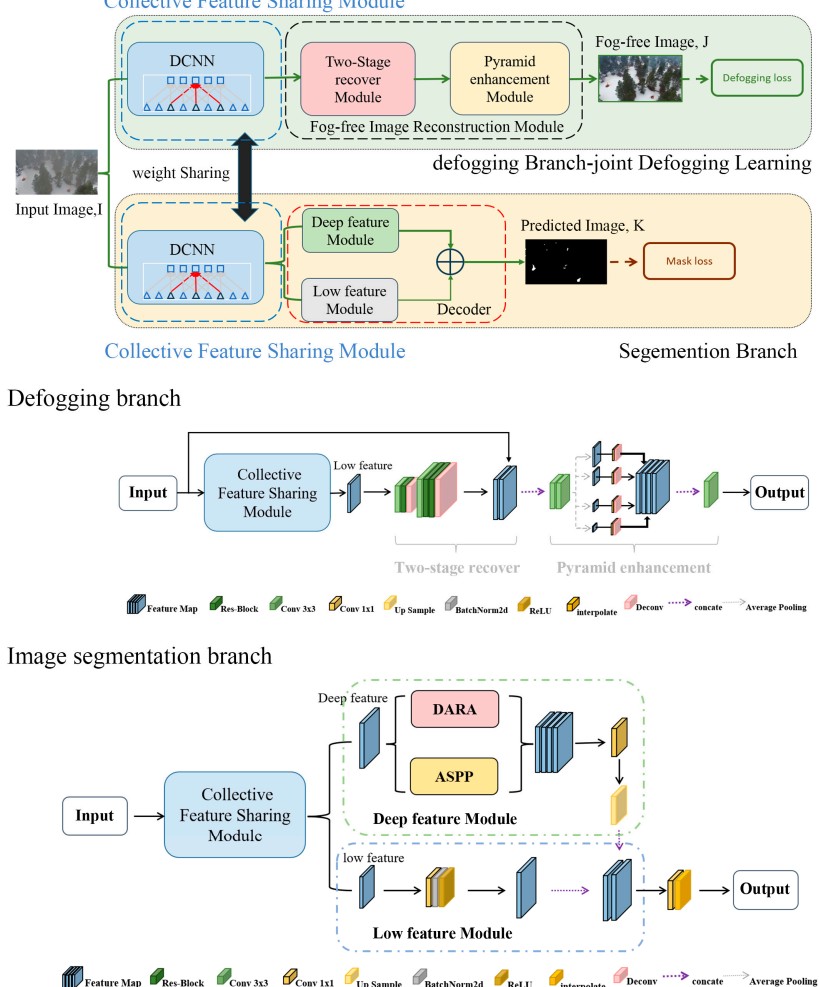

**Figure 4.** The architecture of our joint dehazing learning network for forest fire segmentation. The network consists of two branches: the forest fire division branch and the fog removal branch. Two branches share collective features simultaneously by Collective Feature Sharing Module.

### 2.3. Defogging Branch

The forest fire segmentation branch performs better in foggy conditions thanks to the defogging branch, which is intended to improve the quality of the common features extracted by the CFSM. This branch uses the CFSM and the module for fog-free picture reconstruction to achieve its goal.

### 2.3.1. Shared Fused Feature Extraction Network

The CFSM is designed for extracting features from the input images that contain crucial information for joint learning of fog removal and forest fire segmentation. To keep the architecture simple and avoid adding computational burden to the network, the CFSM was designed based on the DCNN part of the module in the DeepLabV3+ network. While the features derived from the deeper layers of the network contain more high-level information, the features retrieved from the shallower levels of the network contain more spatial and low-level information that assists in the defogging process [20]. As a result, the segmentation branch of the proposed CFSM has shallow convolutional layers. The fog-free picture reconstruction module receives the characteristics that the CFSM retrieved after performing defogging.

### 2.3.2. Fog-Free Image Reconstruction Module

The features extracted by the CFSM called FC may be degraded by fog, resulting in a limited performance of forest fire segmentation. To reconstruct the FC shared with the segmentation branch during joint learning, we propose the fog-free image reconstruction module FIRM, which is composed of the following architecture. First, the extracted features FC are subjected to a convolutional and two residual blocks in order to obtain more precise characteristics. Secondly, a deconvolution operation is carried out twice to upsample these features to match the input resolution as the dimensionality of these features is lowered in the preceding layers. The upsampled features are then concatenated with the input image and passed through the pyramid enhancement module (PEM) to produce the final, fog-free outcome. The PEM module can extend the network's characterization capability by extracting features based on many learning scales and various receptive areas. Our operation is based on pyramid pooling [44].

### 2.4. Image Segmentation Branch

For the forest fire segmentation branch, we utilized DeepLabV3+ as the backbone, with the DCNN shallow convolutional layer serving as the CFSM that is used for feature extraction by the defogging branch. However, the segmentation performance is suboptimal because the original ASPP module in DeepLabV3+ cannot obtain sufficient contextual information when dealing with smaller targets. To overcome this challenge, we designed a novel attention mechanism called DARA, which was connected in parallel to the original ASPP module. This design greatly enhanced the performance of our product.

### 2.4.1. Dual-Attention Residual Module

Our dual-fusion attention residual feature attention mechanism (DARA) first passes the depth features through two dilated convolutions to create new incremental depth features with different centers of gravity in the position attention [45] and channel attention [46] modules.

Obtaining a distinguishing feature representation is crucial for semantic segmentation, and that may be performed by accumulating long-term contextual data [36]. The location–attention module encodes rich contextual information into local features, which are represented as the correlation influence matrix between pixel points, thereby improving the local feature representation. The location attention module's and the computational equations' final output features are listed as follows:

$$S_{ji} = \frac{\exp(B_i^T \cdot C_j)}{\sum\limits_{i=1}^{N} \exp(B_i^T \cdot C_j)} (S \in R^{N \times N}) \tag{5}$$

$$E_j^1 = \alpha \sum_{i=1}^{N} S_{ji} D_i + A_j (E^1 \in R^{C \times H \times W}) \tag{6}$$

where is the deeper feature $\{B, C\} \in R^{C \times N}$, $D \in R^{C \times H \times W}$ obtained by convolving the input deep feature; $S_{ji}$ is a measurement of how the pixel at the place $I$ affects position j and $N = H \times W$.

In computer vision, high-level semantic features can be represented as channel graphs, which are specific to each object class. However, these semantic responses are not independent of each other. To address this, we suggest a channel attention module that simulates the relationships between different channels. This module uses a channel attention feature map to represent the relationships between different semantic responses. The final output features of the channel attention module are denoted as $X \in R^{C \times C}$ and can be computed using the following equations:

$$X_{ji} = \frac{\exp(A_i \cdot A_j)}{\sum\limits_{i=1}^{C} \exp(A_i \cdot A_j)} \tag{7}$$

$$E_j^2 = \beta \sum_{i=1}^{C} (X_{ji} Ai) + Aj \tag{8}$$

where the deeper features $A \in R^{C \times H \times W}$ obtained by the convolution of the input depth features $X_{ji}$ represent the influence of channel $I$ on channel $j$. Our final feature map $E \in R^{C \times H \times W}$ is described as:

$$E_j = E_j^1 + E_j^2 + A_j \tag{9}$$

To improve the characterization of raw features in a semantic segmentation task, we propose to use both raw depth features and learned depth features. We fuse these features and incorporate them into the DeepLabV3+ network architecture. Specifically, we connect the fused depth features to the encoder in parallel with the ASPP module, which improves the network's capacity for feature identification. Additionally, this inclusion of depth features enriches the long-term contextual semantic information captured by the network. Figure 5 shows the enhanced network design.

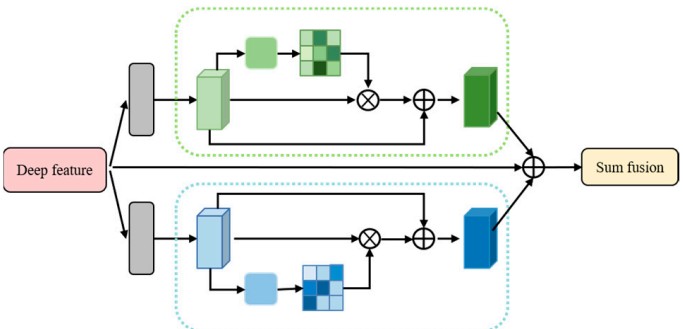

**Figure 5.** DARA: fusion attention residual module.

2.4.2. Loss Function

In the training stage, the loss function curve gives a graphic representation of the model's convergence. For Defog DeepLabV3+, the total loss is composed of two aspects. One is the loss of the defogging branch, which removes the fog from foggy images. The other is the loss generated by the segmentation branch, which splits the input image into patches for processing. The total loss can be expressed as the sum of these two components. Specifically, the mathematical equation for the total loss is:

$$\mathcal{L}_{total} = \mathcal{L}_{DF} + \mathcal{L}_{SG} \tag{10}$$

The aim of minimizing this loss during training is to improve the performance of the Defog DeepLabV3+ model on the task of defogging. The defogging branch's loss function may be represented as follows:

$$\mathcal{L}_{DF} = \frac{1}{Q}\sum_{i=1}^{Q}\left\|J_i - J_i^{GT}\right\|_2 \tag{11}$$

where $\|\cdot\|_2$ denotes the l2 parametric number. $J_i$ and $J_i^{GT}$ denote the predicted fog-free image I and the corresponding ground truth values in a batch, respectively.

The split branch's loss function may be stated as follows:

$$\mathcal{L}_{SG} = \frac{1}{N}\sum_{j}^{n}\left[y_j \ln G_\alpha(X_j|Z) + (1 - y_j)\ln(1 - G_\alpha(X_j|Z))\right] \tag{12}$$

where $G_\alpha(X_j|Z)$ is the label probability with ai pixel $j$ and is the GT label.

### 2.5. The Process of Working with the Model

During the model training phase, it is necessary to run both the dehazing branch and the image segmentation branch simultaneously. However, during the forest fire detection phase, the model only needs to run the image segmentation branch. This working mode ensures the quality of the dehazing segmentation while significantly improving the efficiency of model operation (Figure 6).

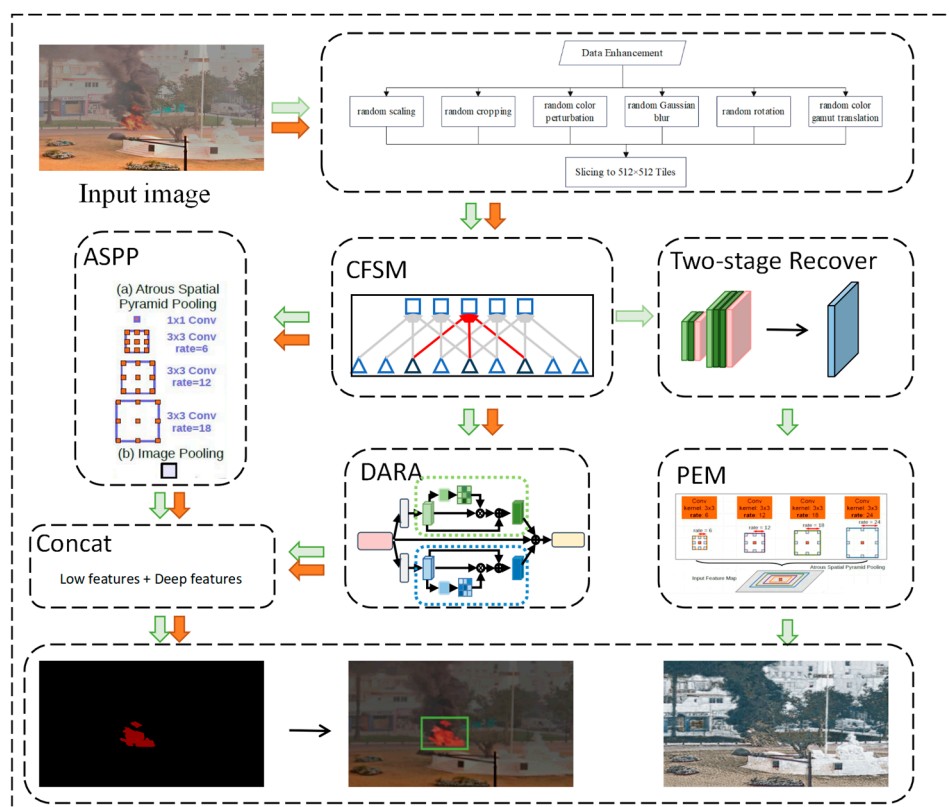

**Figure 6.** The flow chart of the detailed work with the whole model.

## 3. Results

As described in this section, we tested our proposed deep learning framework for defogging and forest fire segmentation in various conditions. The hardware consists of an Intel(R) Xeon(R) Gold 6330 CPU and an NVIDIA RTX 3090 GPU with 24 GB of RAM (Intel and NVIDIA Corporation, Santa Clara, CA, USA).

### 3.1. Defogging Branch

#### 3.1.1. Evaluation of the Effectiveness of Defogging

To verify the usefulness of our defogging module in boosting our model's segmentation performance, we conducted the experiment comparing our Defog DeepLabV3+ to a baseline segmentation model called BaseLine, which was trained without the defogging branch. We used two different backbone models, mobilenetv2 [38] and xception [39] in our experiments. Furthermore, we tested the impact of two additional modules on our model's performance: the DARA module and the PEM module. To ensure fairness in our experiment, we used mean pixel accuracy (mPA) as the evaluation metric. The total number of properly categorized pixels is divided by the total number of pixels in the image to compute mPA.

$$PA = \frac{\sum\limits_{i=0}^{k} p_{ii}}{\sum\limits_{i=0}^{k}\sum\limits_{j=0}^{k} p_{ij}} \tag{13}$$

where $P_{ij}$ Indicates the jth pixel of row. This metric is calculated by dividing the total number of pixels in the image by the number of pixels that have been properly classified.

In Table 3 and Figure 7, our experiment shows that incorporating the defogging branch into the deep learning framework can greatly enhance segmentation performance, especially for the mobilenetv2+ architecture. By adding the defogging branch, the mPA increases to 93.33%. Furthermore, incorporating the pyramid enhancement module PEM into the defogging branch further improves the mPA to 94.26%. These results highlight the effectiveness of our framework in improving segmentation accuracy, especially in challenging foggy environments.

**Table 3.** The efficacy of the suggested cooperative defogging learning method. "BaseLine" indicates that the improved presentation of DeepLabV3+ does not have joint defogging learning. DARA and PEM indicate whether to use the corresponding module. We use the sign '$\sqrt{}$' to indicate whether to enable them and display the highest value of the mPA in bold.

| Module | DCNN | DARA | PEM | mPA (%) |
|---|---|---|---|---|
| Baseline | xception | | | 89.95 |
| BaseLine | mobilenetv2 | | | 91.27 |
| Defog DeepLabV3+ | xception | $\sqrt{}$ | | 92.16 |
| Defog DeepLabV3+ | xception | $\sqrt{}$ | $\sqrt{}$ | 93.18 |
| Defog DeepLabV3+ | mobilenetv2 | $\sqrt{}$ | | 93.33 |
| Defog DeepLabV3+ | mobilenetv2 | $\sqrt{}$ | $\sqrt{}$ | **94.26** |

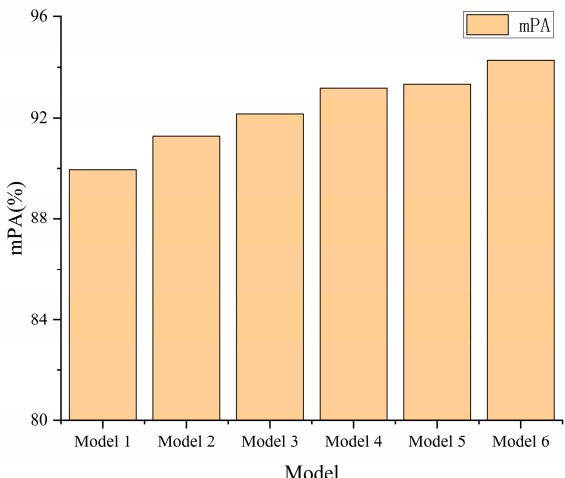

**Figure 7.** The efficacy of the suggested cooperative defogging learning method. Model 1~6, respectively, represent the 6 models from top to bottom in Table 3.

3.1.2. Visualization Analysis

To gain a more intuitive understanding of the effectiveness of our defogging module, we selected the first five images from our dataset to showcase the defogging effect of our model under different scenarios, fog concentrations, and fire sizes. For each image, we arranged the fog map, defogging effect map, and original map for reference.

Figure 8 presents the results of our experiment, which proves that the defogging module can effectively adapt to the changing scenarios of forest fire defogging. Our model can clearly reveal the flames of various shapes and sizes from the fog, and the comparison with the original image still shows strong recognition. These results highlight the ability of our proposed framework to improve the visibility of foggy images.

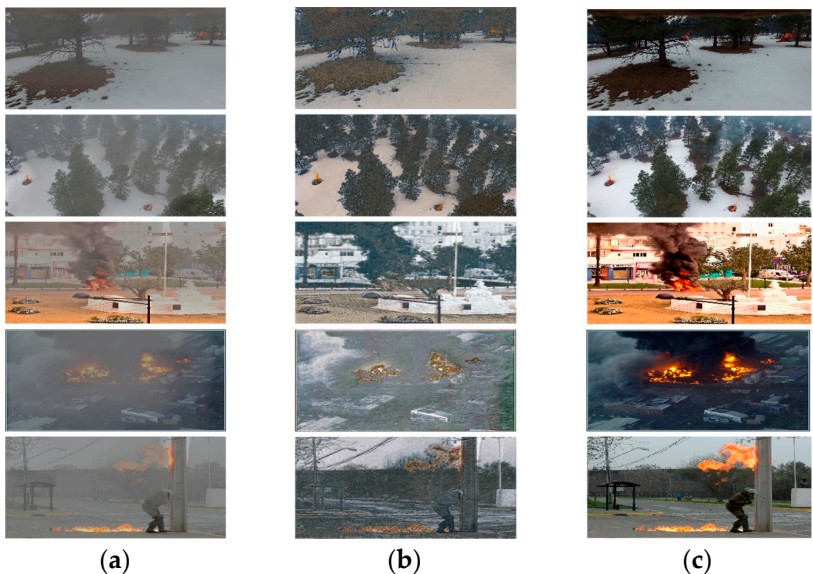

(a)            (b)            (c)

**Figure 8.** Visualization on the FFLAD dataset. (**a**) the fogged image; (**b**) the defogged affected image; and (**c**) the clear image.

*3.2. Forest Fire Segmentation*

3.2.1. Assessment Indicators

In addition to the mean pixel accuracy (mPA) metric, we also evaluated the performance of our Defog DeepLabV3+ model using the intersection over union (IoU) metric. This statistic calculates the percentage of overlap between anticipated segmentation results and their related ground truth labels. A value closer to 1 indicates better segmentation performance.

To ensure fairness in our evaluation, we computed the mean IoU to quantify our model. The equation for computing mean IoU is as follows:

$$mIoU = \frac{1}{N}\sum_{i=1}^{N}\frac{P_i \cap G_i}{P_i \cup G_i} \tag{14}$$

where $P_i$ and $G_i$ denote the predicted result of the $i$-th image and the corresponding ground truth label, respectively. This metric provides a more comprehensive evaluation of our model's segmentation performance, which is critical for applications like forest fire detection and prevention.

In our experiments, the target is considered a positive sample if the IoU is 0.5 or above, and a negative otherwise. In addition, an additional metric from recall can be used, denoted as:

$$Recall = \frac{TP}{TP + FP} \tag{15}$$

where *TP* represents true positive cases and *FN* represents false negative cases. It represents the ratio of right forecasts to total positive case predictions.

To ensure fairness in our evaluation, we computed the mean recall on the test set as another measure of our model's performance. The equation for computing mean recall is similar to that of mean IoU. This metric provides a complementary evaluation of our model's segmentation performance, as it focuses on the ability of our model to correctly identify positive cases.

### 3.2.2. Performance Assessment

The enhanced forest fire semantic segmentation model, built upon DeepLabV3+, integrates defogging optimization to generate defogging maps and forest fire segmentation simultaneously. In Figure 9, the green box highlights the area with flames, which is the target for segmentation. Image a shows the original input images used in the foggy forest fire segmentation task. The experimental results highlighted in red in Image b demonstrate the commendable performance of the model. When compared with the ground truth labels in Figure 9c, we observe that our model can accurately capture small, hidden small flames, such as flame areas 2, 3, 5 and 6. The flame edge segmentation is also very detailed, such as flame areas 1, 4, 7 and 8.

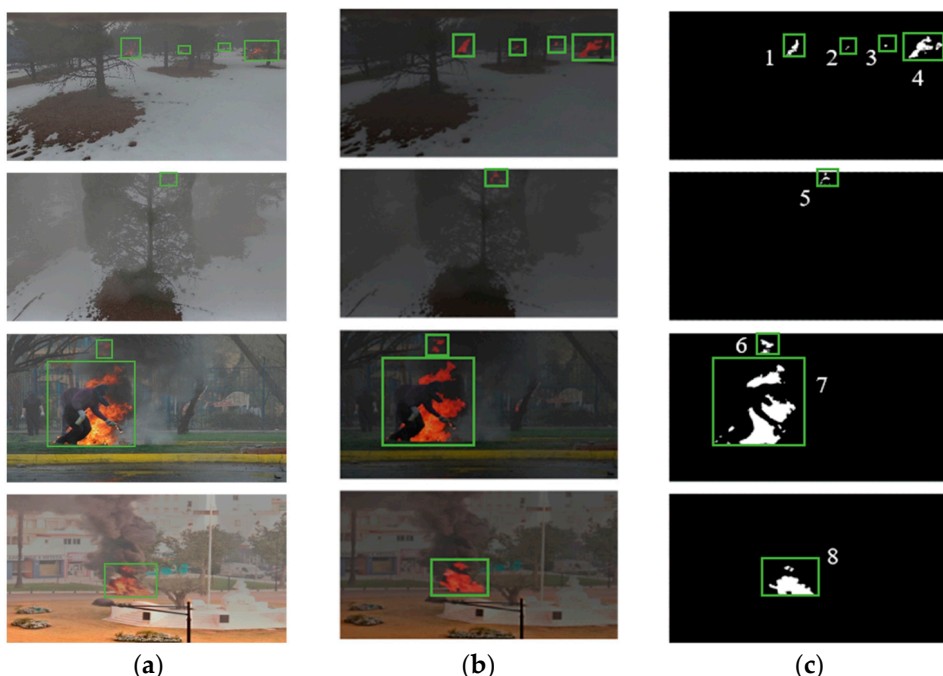

**Figure 9.** Results of Defog DeepLabV3+ for forest fire segmentation. All flame areas are highlighted with a green box with a serial number. (**a**) Raw images; (**b**) predicted images; and (**c**) ground truth.

In the visual representation of images, flames have the characteristics of high luminance and high contrast [47]. This is similar to lamps, the sun, etc. Our model should be able to accurately identify flames and distinguish these flame-like objects. In Figure 10, Image a from top to bottom is lanterns, the sun, the chandelier, and the fire hydrant. Although our model has a very small amount of misjudged pixels in Image b, such as area 1, 2 and 3, the similar objects are well distinguished from the flames. Specifically, our model achieves a high precision of 99.7% for flame segmentation, demonstrating superior performance in distinguishing flames from similar objects.

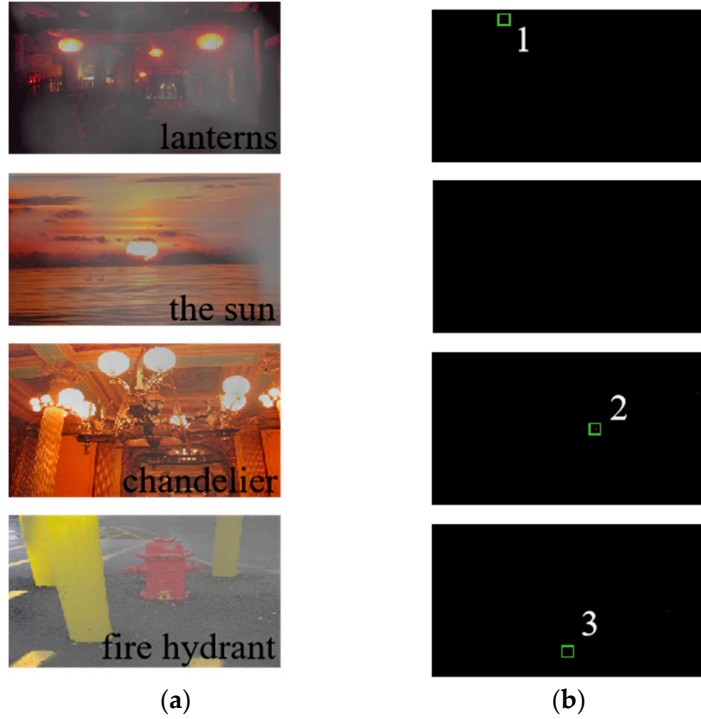

**Figure 10.** Flame discrimination experiment in other environments involving other similar object types. (**a**) Raw image with other similar objects; (**b**) predicted result. Flame recognition areas are highlighted with a green box with a serial number.

Our experiments used five-fold cross-validation, which is consistent with the data splitting ratio we agreed upon. First, the dataset was randomly divided into five equal-sized folds. Then, for each fold i, the i-th fold was used as the test set, and the other four folds were used as the training set. In this way, five models were obtained, and each model used different training and test sets. Table 4 and Figure 11 present the experimental findings. All models achieved high performance in terms of mPA, mIoU, and mRecall, with an average performance of 94.23%, 89.51%, and 94.04%, respectively. Furthermore, the model's performance on the two test sets was not much different. This demonstrates the strong generalization performance of our model.

**Table 4.** The result of 5-fold cross-validation. Test1 represents the split test set and Test2 represents the additional test set.

| Model | Test Set | mPA (%) | mIoU (%) | mRecall (%) |
|-------|----------|---------|----------|-------------|
| Model 1 | Test 1 | 94.10 | 89.32 | 93.98 |
|         | Test 2 | 94.00 | 89.25 | 94.03 |
| Model 2 | Test 1 | 94.22 | 89.48 | 94.15 |
|         | Test 2 | 94.12 | 89.43 | 94.16 |
| Model 3 | Test 1 | 94.35 | 89.67 | 94.27 |
|         | Test 2 | 94.25 | 89.62 | 94.29 |
| Model 4 | Test 1 | 94.18 | 89.42 | 94.08 |
|         | Test 2 | 94.08 | 89.37 | 94.10 |
| Model 5 | Test 1 | 94.30 | 89.61 | 94.17 |
|         | Test 2 | 94.20 | 89.56 | 94.19 |
| Average | Test 1 | 94.23 | 89.51 | 94.04 |
|         | Test 2 | 94.13 | 89.45 | 94.15 |

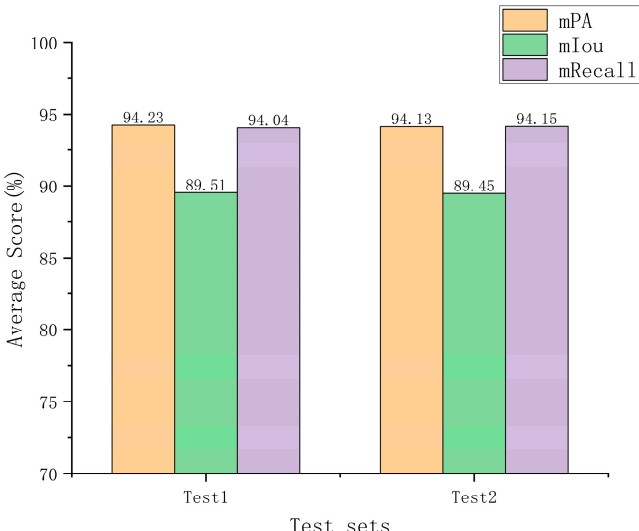

**Figure 11.** The average model performance of 5-fold cross-validation. Test1 represents the split test set and Test2 represents the additional test set.

The confusion matrix for the model 1 test set is shown in Figure 12, and the ROC curve drawn by moving the threshold is shown in Figure 13, with the AUC calculated as 0.97 (±0.015) according to the curve. The model performs well with strong classification ability.

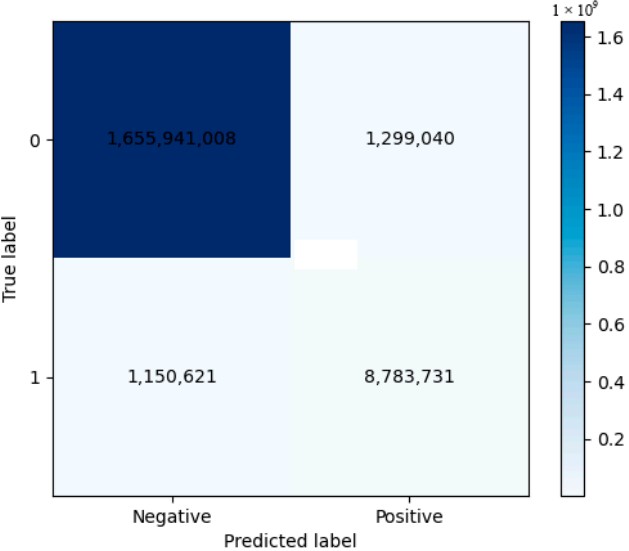

**Figure 12.** The confusion matrix.

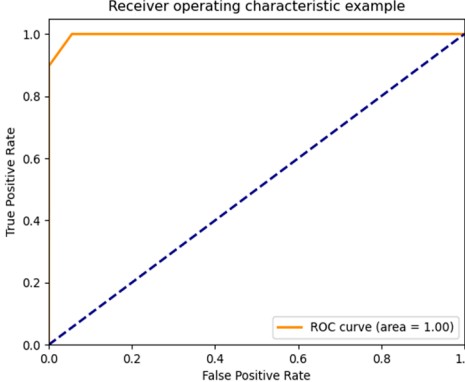

**Figure 13.** The ROC curve. The blue line represents the result of a random guess.

Furthermore, Figure 14 shows the loss curves of our model during training for 100 epochs on the training and validation sets. It reflects that in model 1, the performance of the test set and the training set change with iterations. The overall smooth and downward trend, and gradually converge after about 80 epochs. This demonstrates how well our model can gradually improve the segmentation results by learning the properties of the input image.

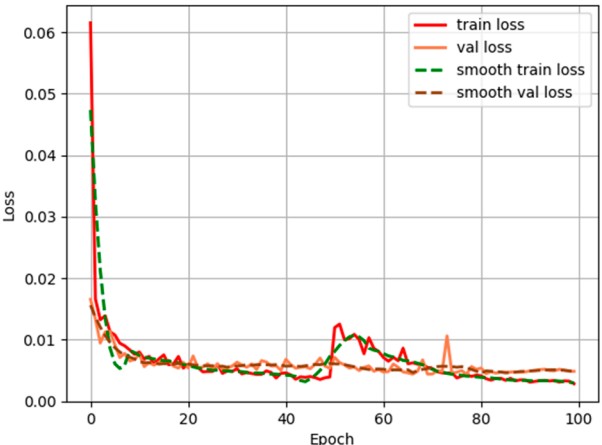

**Figure 14.** Loss curves of Defog DeepLabV3+.

To prove the superior performance of the Defog DeepLabV3+ model for forest fire segmentation, we compared our model with many cutting-edge deep-learning-based models, such as UNet, PSPNet [42], and DeepLabV3+. We used the FFLAD dataset and configuration to train all models to guarantee it to be persuasive.

We chose four representative images and display them in Figure 15. Most models produced relatively accurate segmentations for obvious targets that were distinguishable from the surrounding area and had low haze. However, for forest fires with small targets or high haze, shown with the green boxes in Figure 15a,b, most models had a little under-segmentation, except UNet-agg, DeepLabv3+, and our Defog DeepLabV3+. For small and hidden flames such as flame areas 2, 3, and 5, PSPNet could not correctly identify them. Image h of flame area 5 shows that DeepLabV3+ is not very fine-grained for flame edges. Both PSPNet and DeepLabV3+ have serious false detections (marked in blue boxes). Our model outperformed other segmentation models, especially on images with large haze and complex scenarios.

Table 5 and Figure 16 present the quantitative analysis of the compared models, where evaluation metrics such as mPA, mIoU, and mRecall were used to assess their performance. Our proposed model achieved the second-highest mPA and mIoU scores among the compared models, while also achieving the highest mRecall score in both of the test sets. These results suggest that the proposed model is able to accurately identify objects in the dataset, while also capturing the overall structure and context of the scene. Overall, the proposed model shows promise as an effective approach for semantic segmentation on this particular dataset. The combination of high mPA, mIoU, and mRecall scores indicates that the proposed model can achieve both accurate and comprehensive segmentation results, making it a valuable tool for applications that require precise object detection and segmentation.

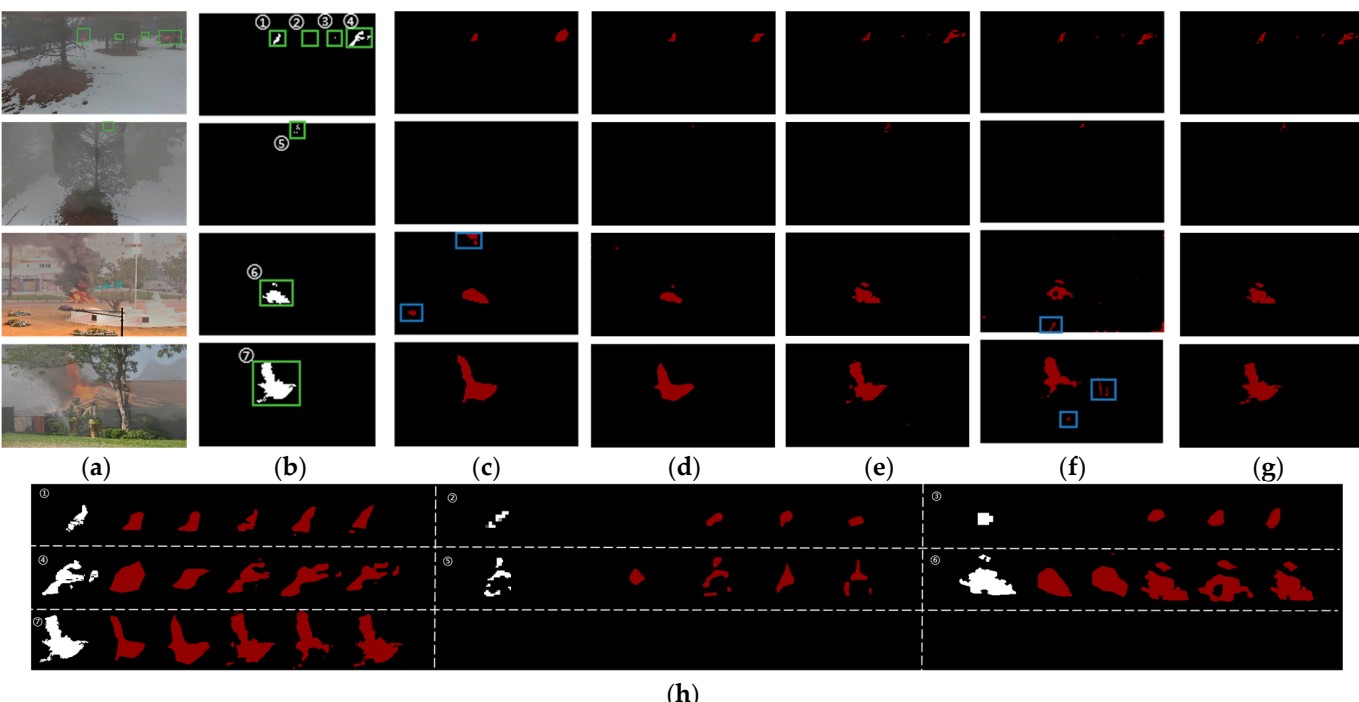

**Figure 15.** Results on testing images. Flame recognition areas are highlighted with a green box with a serial number. Flame segmentation details are highlighted with blue boxes. (**a**) Raw images; (**b**) ground truth. Segmented images by (**c**) PSPNet-mobilenet, (**d**) PSPNet-resnet50, (**e**) UNet-vgg, (**f**) DeepLabv3+, (**g**) our Defog DeepLabV3+, and (**h**) segmentation details. Numbers (1~7) represent the order of fire points, and from left to right are the predicted results generated by the corresponding ground truth, PSPNet-mobilenet, PSPNet-resnet50, UNet-vgg, DeepLabv3+, and our Defog DeepLabV3+ label.

**Table 5.** Comparison of different deep learning segmentation methods on FFLAD. Test1 represents the split test set and Test2 represents the additional test set.

| Methodology | Test Set | mPA (%) | mIoU (%) | mRecall (%) |
|---|---|---|---|---|
| PSPNet-mobilenetv2 | Test 1 | 83.15 | 71.64 | 77.93 |
| | Test 2 | 83.10 | 71.55 | 77.85 |
| PSPNet-resnet50 | Test 1 | 80.70 | 75.48 | 87.68 |
| | Test 2 | 80.80 | 75.55 | 87.75 |
| UNet-vgg | Test 1 | 95.00 | 88.95 | 92.86 |
| | Test 2 | 95.20 | 89.12 | 92.96 |
| DeepLabV3+-mobilenetv2 | Test 1 | 91.27 | 84.29 | 90.15 |
| | Test 2 | 91.40 | 84.48 | 90.33 |
| Ours | Test 1 | 94.26 | 89.51 | 94.04 |
| | Test 2 | 94.20 | 89.49 | 94.02 |

To demonstrate the importance of introducing the attention mechanism in the segmentation branch of our model, we conducted a simple but necessary ablation experiment. Table 6 displays the experiment's outcomes, and they indicate that the introduction of the Dense Attention Refinement Module (DARA) significantly improves the mean intersection over union (mIoU) by about 1.5% in both cases. And Figure 17 fully demonstrates the superiority of Model 3. This suggests that by carefully exploiting the inherent information of the features, our attention strategy may improve model accuracy.

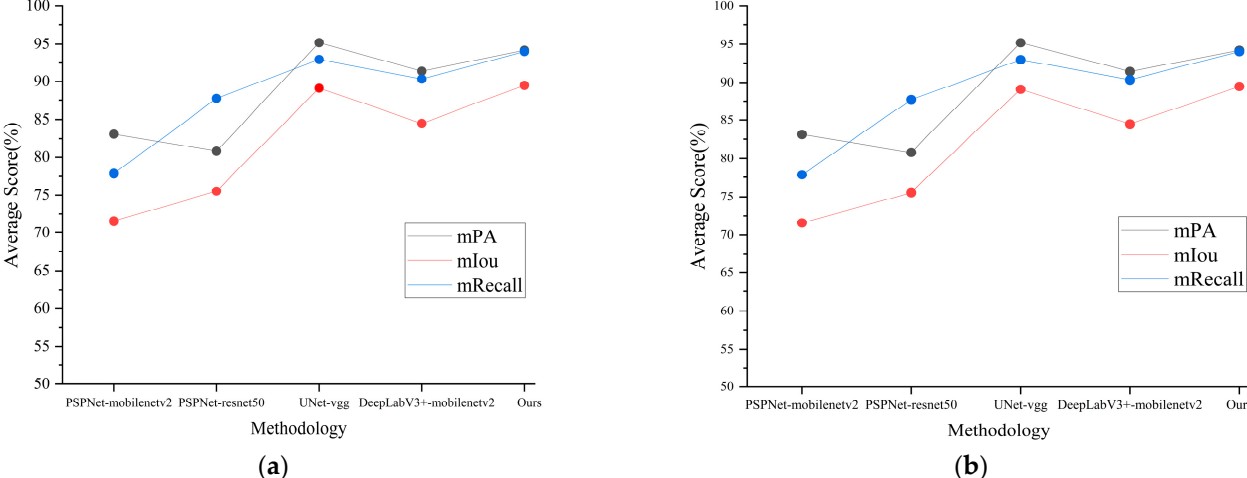

(**a**)  (**b**)

**Figure 16.** Comparison of different deep learning segmentation methods on FFLAD. (**a**) The split test set and (**b**) the additional test set.

**Table 6.** Ablation experiment for the DARA module. We use the sign '$\surd$' to indicate whether to enable the DARA module.

| Model | DCNN | DARA | mPA (%) | mIoU (%) | mRecall (%) |
|---|---|---|---|---|---|
| Defog DeepLabV3+ | xception | $\surd$ | 93.18 | 88.05 | 93.03 |
| Defog DeepLabV3+ | xception | | 92.10 | 86.72 | 92.61 |
| Defog DeepLabV3+ | mobilenetv2 | $\surd$ | 94.26 | 89.51 | 94.04 |
| Defog DeepLabV3+ | mobilenetv2 | | 93.30 | 88.00 | 93.09 |

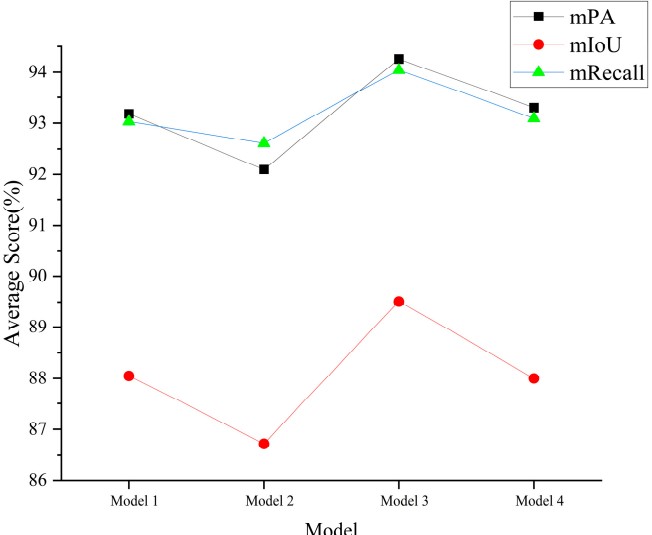

**Figure 17.** Ablation comparison for the DARA module. All models are based on Defog DeepLabV3+. Model 1 and 2 represent xception with DARA or not. Models 3 and 4 represent mobilenetv2 with DARA or not.

In addition, we analyzed the importance of different pixel features after adding the DARA module through a heatmap. As shown in the Figure 18, Figure 18b was generated by the model and it shows the result of combining the feature map with the original image. The color of the heatmap represents the importance of different regions of the image to the model's judgment. In this image, we can see that some areas are red or yellow, which means that these areas play an important role in the model and have a greater impact on the final prediction results. And some areas appear blue, which means that these areas

have less influence on the final prediction results. The flame feature is more important than other parts in our model. This suggests that the DARA module can effectively capture the flame features. This analysis provides valuable insights for further improving the model and optimizing the segmentation results.

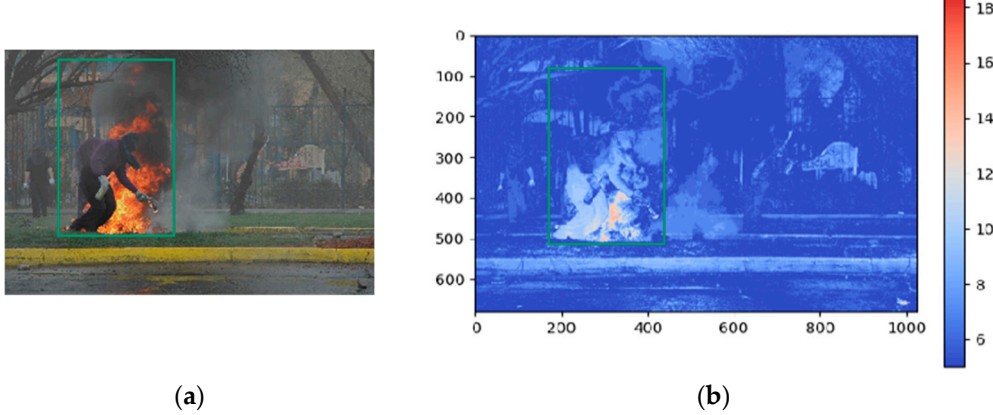

**Figure 18.** Heat map analysis of image features. Flame recognition areas are highlighted with a green box. (**a**) The original image; (**b**) the heat map of the fusion of the intermediate feature map and the original image.

We estimated the model size and inference time of our proposed framework in addition to analyzing the segmentation performance of the comparable approaches. Compared to the original DeepLabV3+ model, our improved model (Defog DeepLabV3+) is slightly larger in terms of the total number of parameters, with an increase of about 46%. This increase is mainly due to the addition of a new defogging branch in our model.

In Table 7 and Figure 19, we can see that despite the increase in model size, the inference time of our improved model-B is almost the same as that of DeepLabV3+. This is because only the computation of the split branch is involved in the inference process. Therefore, the modest increase in computational overhead is worth the improvement in segmentation performance achieved by our proposed framework.

**Table 7.** Model running efficiency comparison.

| Module | DCNN | Total Number of Parameters | Inference Time (ms) |
|---|---|---|---|
| DeepLabV3+ | xception | 9,827,466 | 3357.82 |
| DeepLabV3+ | mobilenetv2 | 5,849,070 | 3016.42 |
| Defog DeepLabV3+ | xception | 12,984,029 | 3829.34 |
| Defog DeepLabV3+ | mobilenetv2 | 9,596,745 | 3339.61 |

The evaluation of both model size and inference time is important for practical applications, as it provides information about the efficiency and feasibility of deploying the model in resource-constrained environments. Our evaluation's findings show that the suggested structure successfully strikes a compromise between computational effectiveness and the efficiency of segmentation. This is essential for real-world applications like forest fire detection and prevention.

Finally, we performed sensitivity analysis on the model input parameters, including the batch size, initial learning rate, and number of iterations. The model performance was evaluated by the loss function on the test set, as shown in Figure 20. As we can see from the figure, the batch size fluctuated around 0.07 with a range of 0.01, the initial learning rate was between 0.07 and 0.15, and the number of iterations converged to 0.073. It can be observed that these three parameters had little sensitivity to the model performance.

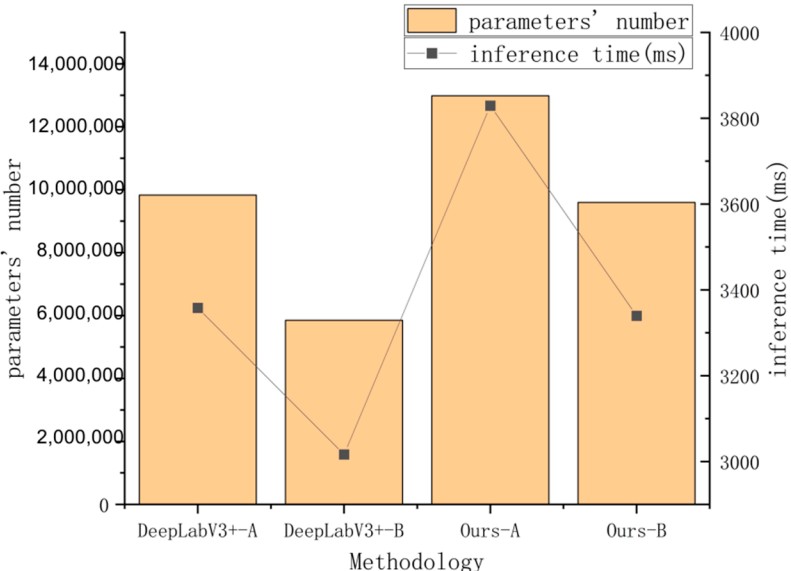

**Figure 19.** Model running efficiency comparison. Alphabet A represents Backbone xception and alphabet B represents Backbone mobilenetv2.

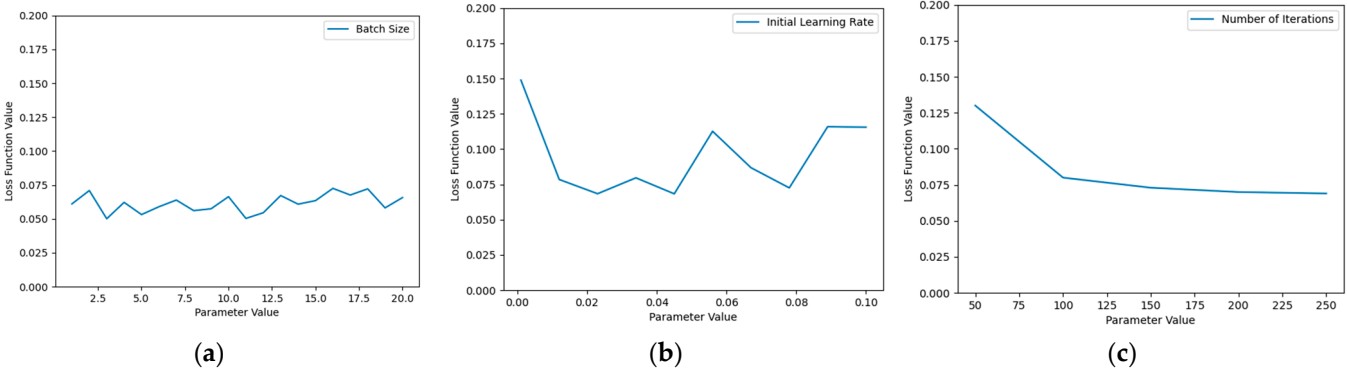

| (a) | (b) | (c) |

**Figure 20.** Sensitivity analysis on the model input parameters. (**a**) The impact of Batch size on the model performance; (**b**) The impact of initial Learning Rate on the model performance; (**c**) The impact of Number of iterations on the model performance.

## 4. Discussion

Forest fires are complex and dynamic events that constantly change over time, making it difficult to accurately capture and describe features such as flames and smoke. The shape of the fire is also altered by the terrain, further complicating the detection and segmentation of forest fires [48]. Additionally, foggy weather reduces visibility and change the color and brightness of images, further challenging the extraction of detailed information from images [11].

To address these challenges, determining the extent of forest fires on foggy days has become an important and challenging problem that needs to be solved. In previous research on defogging semantic segmentation, many methods have adopted a two-stage processing strategy, where defogging image extraction is followed by semantic segmentation [24,25]. However, this approach may result in information loss, as important semantic information may be lost during the defogging image extraction process. Furthermore, the two-stage processing strategy requires the application of two models in the inference stage, which may require more data and computational resources.

In our study, we propose a new joint defogging method that improves the existing semantic segmentation model. Our approach combines two tasks of defogging with forest fire segmentation and employs a smart attention mechanism (DARA) to improve the

display of feature channels. We investigated clean basic information and deep complex meanings to accomplish accurate forest fire segmentation. To assess the efficacy of our suggested technique, we first examined the joint defogging branch and showed that our approach was significantly effective. We then compared our model with existing state-of-the-art semantic segmentation models, which use traditional defogging methods. Our results demonstrate that our model outperformed the comparison methods in both recall and mIoU metrics, indicating that our model had good performance in solving the forest fire detection problem in foggy weather.

We also examined the effectiveness of the DARA attention module and showed that it was effective in extracting depth image features. Additionally, we compared our proposed Defog DeepLabV3+ model with the original DeepLabV3+ model and showed that our method delivered a modest improvement with an acceptable increase in overhead.

Compared to the previous multi-stage processing strategies adopted by researchers, such as the GXTD detection model with defogging capabilities, the GXTD detection model improves image quality and fire detection accuracy by removing haze from images [23]. However, this approach requires additional computational cost, which may result in higher execution time, especially when processing a large number of images. On the other hand, the smoke detection model that utilizes dark channel-aided mixed attention does not require additional computational cost, but the performance of the model is much inferior to ours due to the prior-based approach [21].

Despite the high accuracy achieved by our Defog DeepLabV3+ model in fogged forest fire segmentation, there are still some shortcomings that need to be addressed. In further research, we plan to streamline the learning module for deep segmentation features and explore semi-supervised training schemes to bridge the domain gap between real and synthetic datasets [27]. Overall, our proposed method shows promise in improving the accuracy and robustness of forest fire detection and prevention, which is crucial for protecting our environment and communities.

## 5. Conclusions

In this study on precise segmentation of forest fires in foggy weather, our main achievements were as follows:

- We proposed the Defog DeepLabV3+ joint defogging and forest fire segmentation model to address foggy weather challenges in detecting forest fires.
- We designed the DARA attention mechanism to enhance feature channel representation and improve forest fire segmentation accuracy.
- We constructed the FFLAD dataset with synthetic and real foggy images to train our model.
- Our experimental results show that Defog DeepLabV3+ outperforms state-of-the-art methods with 94.26% accuracy, 94.04% recall, and 89.51% mIoU.

Each component of our proposed model contributes to the overall performance, indicating the potential to improve forest fire detection and prevention in foggy weather. This is crucial for environmental and community protection.

**Author Contributions:** T.L. devised the programs and drafted the initial manuscript. W.C. contributed to the writing and the experiments. X.L. and Y.M. helped with the data collection and data analysis. J.H. helped to improve the manuscript and modified the model in the later stage. D.G. revised the initial manuscript. J.X. designed the project and revised the manuscript. All authors have read and agreed to the published version of the manuscript.

**Funding:** This work was supported by the Natural Science Foundation of Jiangsu Province (BK20211357), the Qing Lan Project of Jiangsu Province.

**Data Availability Statement:** Not applicable.

**Conflicts of Interest:** The authors declare no conflict of interest.

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
