# Peer review of "Defogging Learning Based on an Improved DeepLabV3+ Model for Accurate Foggy Forest Fire Segmentation"

_forests, doi:10.3390/f14091859_

Round 1
Reviewer 1 Report
My review comments are attached.

Editing of the English language is required
Reviewer 2 Report
Dear Authors,
Deep learning technology managed to develop strategies for detecting forest fires. Nonetheless, accurately detecting and segmenting forest fires under different conditions is challenging, particularly in foggy environments. This manuscript proposed a collaborative defogging learning framework according to the improved DeepLabV3+ simulation. The manuscript analyzed a novel matter in the field of fire that can be helpful for researchers. However, the authors should consider some minor revisions before publications.
-Abstract should be useful and brief. It is recommended to rewrite the abstract. It should show the overview of the manuscript. Do not use active sentences.
-In the section of 2.1.1, it is recommended to use further data source to obtain more accurate outcomes.
-There are some English language errors that should be addressed.
-Recheck equation (5). Possibly there is a missing parameter.
-Conclusion should show the main achievements of the manuscript. It is recommended to use bullet points to show them.
Regards
There are some English language errors that should be addressed.
Round 2
Reviewer 1 Report
Dear Authors,
-Figures 12 and 17 are not readable.
-Figures 3 to 8, too many descriptive flowcharts. make one or two telling the main point
-For Figures 10, 11, 16, and 18, you need to write a note on each image and tell the main outcome of it
-For visualization figure, it should come with a plot
-57 references will reduce the novelty of the work.
it needs moderate check
